# Dopamine maintains network synchrony via direct modulation of gap junctions in the crustacean cardiac ganglion

Brian J Lane[1][†], Daniel R Kick[1], David K Wilson[1], Satish S Nair[2], David J Schulz[1]*

[1]Division of Biological Sciences, University of Missouri, Columbia, United States; [2]Department of Electrical Engineering and Computer Science, University of Missouri, Columbia, United States

**Abstract** The Large Cell (LC) motor neurons of the crab cardiac ganglion have variable membrane conductance magnitudes even within the same individual, yet produce identical synchronized activity in the intact network. In a previous study we blocked a subset of $K^+$ conductances across LCs, resulting in loss of synchronous activity (Lane et al., 2016). In this study, we hypothesized that this same variability of conductances makes LCs vulnerable to desynchronization during neuromodulation. We exposed the LCs to serotonin (5HT) and dopamine (DA) while recording simultaneously from multiple LCs. Both amines had distinct excitatory effects on LC output, but only 5HT caused desynchronized output. We further determined that DA rapidly increased gap junctional conductance. Co-application of both amines induced 5HT-like output, but waveforms remained synchronized. Furthermore, DA prevented desynchronization induced by the $K^+$ channel blocker tetraethylammonium (TEA), suggesting that dopaminergic modulation of electrical coupling plays a protective role in maintaining network synchrony.
DOI: https://doi.org/10.7554/eLife.39368.001

*For correspondence:
SchulzD@missouri.edu

Present address: [†]Department of Biology, Brandeis University, Waltham, United States

Competing interests: The authors declare that no competing interests exist.

## Introduction

Neural networks must be capable of producing output that is robust and reliable, yet also flexible enough to meet changing environmental demands. One mechanism of providing flexibility to network activity is neuromodulation, which reconfigures network output by altering a subset of cellular and synaptic conductances (*Harris-Warrick, 2011*; *Bargmann, 2012*; *Daur et al., 2016*). However, many networks achieve stable output by a variety of solutions; intrinsic membrane conductances and synaptic strengths can be highly variable yet still produce nearly identical physiological activity (*Ball et al., 2010*; *Calabrese et al., 2011*; *Marder, 2011*; *Ransdell et al., 2013a*). This raises a fundamental question about neuromodulation, highlighted in a recent review by *Marder et al. (2014)*, as to whether modulation of networks with variable underlying parameters can produce predictable and reliable results. These authors demonstrate computationally that modulation of neurons with similar outputs arising from variable underlying conductances can cause anywhere from relatively small to fairly substantial differences in output (*Marder et al., 2014*). Therefore, the response of any neural network to modulation is likely state-dependent (*Goldman et al., 2001*; *Nadim et al., 2008*; *Gutierrez et al., 2013*; *Williams et al., 2013*; *Marder et al., 2014*), and potentially unpredictable as a result of these varying underlying conductances (*Marder et al., 2014*). In some cases neuromodulation can expand the parameter space in which a given activity feature is maintained (*Grashow et al., 2009*), potentially leading to protective effects of modulation that ensure robust network output (*Städele et al., 2015*). Yet these questions have never been addressed in a network that relies on synchronous activity for appropriate physiological output.

The crustacean cardiac ganglion (CG) is a central pattern generator network that produces rhythmic bursts with precisely synchronized activity across all five Large Cell (LC) motor neurons (*Lane et al., 2016*; *Figure 1*). Despite virtually identical output across LCs within a given network, constituent LCs are variable in many ionic conductances, including A-type K$^+$ ($I_A$), high-threshold K$^+$ ($I_{HTK}$), and voltage-dependent Ca$^{2+}$ ($I_{Ca}$) (*Ransdell et al., 2013a*; *Ransdell et al., 2013b*; *Lane et al., 2016*). When subsets of K$^+$ conductances are blocked in LCs of a network, synchrony is disrupted, although ultimately is restored by a combination of compensatory changes in membrane conductance and electrical synaptic strength (*Lane et al., 2016*). The CG is modulated by many substances, including neuropeptides and the biogenic amines serotonin and dopamine (DA) (*Cooke, 2002*; *Cruz-Bermúdez and Marder, 2007*) that are known to target the same K$^+$ conductances that lead to desynchronization when altered (*Kloppenburg et al., 1999*; *Peck et al., 2001*; *Johnson et al., 2003*; *Gruhn et al., 2005*). Therefore, one potentially detrimental impact of neuromodulation altering membrane conductance is a resulting loss of LC synchrony. Given that hormonal modulators in the hemolymph will bathe LCs uniformly, this study addresses whether the CG is tuned to maintain stable synchrony during neuromodulation or if altering a subset of cellular conductances with neuromodulation will desynchronize network activity.

We hypothesized that neuromodulation will desynchronize LC activity owing to the variable conductances across LCs. We tested this hypothesis by exposing the CG to two amine modulators, serotonin (5HT) and DA, and measuring the effects of the modulators on excitability and synchrony individually and when co-applied. We found that serotonergic modulation desynchronizes LC voltage waveforms, and in most networks elicited a distinct mode of output characterized by prolonged pacemaker bursts which drive two distinct LC bursts before the cycle was reset. In contrast, DA had an overall excitatory effect on LCs, but neither desynchronized LC activity nor elicited the two LC bursts per pacemaker cycle seen in 5HT. When co-applied, DA prevented the 5HT-induced desynchronization without preventing the characteristic 5HT output with two bursts per pacemaker cycle. Even with $I_{HTK}$ reduced by TEA application – a known perturbation that leads to substantial loss of

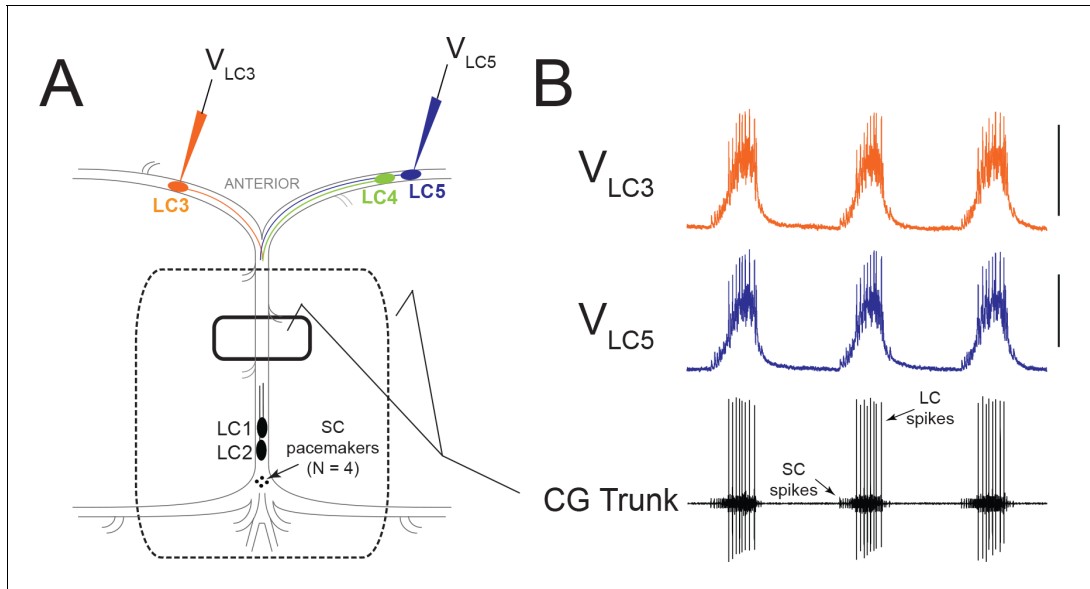

**Figure 1.** Experimental Setup and Typical Activity in the CG. (**A**) Intracellular electrodes recorded simultaneously from LC3 and either LC4 or LC5. Extracellular recordings were taken from a petroleum jelly well on the CG trunk (solid line). For experiments which applied modulators or TEA exclusively to anterior LCs, the extracellular recording was taken from a single large well allowing anterior LCs to be exposed to the perfusate while protecting the remainder of the ganglion (dashed line). (**B**) Representative control activity showing the rhythmic synchronized bursting of LCs, with small cell (SC) pacemaker and Large Cell (LC) motor neuron spikes labeled on the extracellular trace (Scale bars = 10 mV, recording length = 14 s).
DOI: https://doi.org/10.7554/eLife.39368.002

synchrony across LCs (*Lane et al., 2016*) – co-application of DA with TEA prevented desynchroniza-tion. Our results suggest that DA preserves network synchrony by directly targeting and increasing electrical synaptic conductance. Thus, DA may function to maintain robust synchrony in the cardiac network while still being permissive to plasticity of output caused by other modulators.

## Results

### 5HT and DA have distinct excitatory effects when applied to the entire network

Both 5HT ($10^{-6}$M) and DA ($10^{-5}$M) are excitatory when applied to the entire CG of *C. borealis* (*Cruz-Bermúdez and Marder, 2007*), and our results recapitulate this same effect (*Figure 2*). 5HT significantly increased pacemaker burst duration, and in 6 out of 8 experiments switched the net-work to a distinct output consisting of a single prolonged pacemaker burst driving two different and distinct LC bursts that we term 'double-bursting' (*Figure 2A*). Because this represents a distinct mode of firing from control, direct comparisons of LC burst characteristics between these two modes of firing did not seem appropriate. Therefore, changes in network output in 5HT were quanti-fied by analysis of phase relationships among SCs and LCs derived from extracellular recordings (*Figure 2B*, N = 8). 5HT significantly altered the SC off phase (p=0.007, paired t-test), leading to prolonged pacemaker bursting. Because double-bursting is a distinct output from the control rhythm, direct comparisons are more difficult to make for LCs. However, in comparing the first LC burst in preparations that transitioned to double-bursting in 5HT (*Figure 2B*; LC$_{DB}$), there was a sig-nificant phase advance in both the LC on (p<0.001, paired t-test) and LC off (p<0.001, paired t-test) relationships relative to control. Phase relationships were not tested for single-bursting preparations in 5HT, as sample size was only two for this output type. However, the phase relationships of the sin-gle-bursting preparations are shown in *Figure 2B* (LC$_{SB}$).

DA also increased network excitability, but in distinct ways from 5HT (*Figure 2C*). DA never induced double-bursting, but significantly increased the number of LC spikes per burst (8.850 ± 3.978 in control; 11.225 ± 5.337 in DA; p<0.05), LC spike frequency (8.854 ± 4.878 in con-trol; 11.268 ± 7.434 in DA; p<0.05), LC burst duration (0.715 ± 0.280 in control; 0.806 ± 0.281 in DA; p<0.05), and LC duty cycle (0.164 ± 0.0744 in control, 0.211 ± 0.0411 in DA; p<0.05, paired t-tests, N = 8). There were no significant changes in phase relationships in DA relative to control (*Figure 2D*).

When applied focally to the anterior LCs at these same concentrations, our results demonstrate that both amines have direct excitatory effects on LCs. 5HT applied only to the anterior LCs *never* resulted in double-bursting, demonstrating that this output requires 5HT modulation of SC pace-makers. Because focal application of 5HT generated single-bursting output, we could compare burst statistics relative to control. Focal application of 5HT increased the number of spikes per burst (5.850 ± 2.72 in control; 10.85 ± 2.96 in 5HT; p<0.01, N = 6), the spike frequency within each burst (6.24 ± 3.18 Hz in control; 11.53 ± 3.39 Hz in 5HT; p<0.01, N = 6), burst duration (0.642 ± 0.367 s in control; 0.739 ± 0.411 s in 5HT; p<0.05, N = 6), LC duty cycle (0.183 ± 0.059 in control; 0.232 ± 0.063 in 5HT; p<0.01, N = 6) and decreased the LC interburst interval (3.15 ± 0.68 in control; 2.761 ± 0.658 in 5HT; p<0.01, N = 6). DA also directly affected LC output, significantly increasing the number of spikes per burst (4.45 ± 3.42 in control and 10.20 ± 6.41 in DA; p<0.05, N = 8) and spike frequency within each burst (5.85 ± 4.95 in control and 10.38 ± 10.16 in DA; p<0.05, N = 8). Although pacemaker cells were not directly modulated, modest effects on pacemaker bursting were sometimes evident along with the onset of increased LC excitability. This is presumably an indirect effect due to strong electrotonic feedback from LCs, which influences pacemaker activity and can modify the timing of pacemaker bursts in the CG (*Berlind, 1989*; *García-Crescioni and Miller, 2011*).

### 5HT desynchronizes burst waveforms but DA does not

The degree of synchrony in burst waveforms was quantified as described previously (*Lane et al., 2016*, see Materials and methods). Briefly, we performed a cross-correlation on the digitized voltage waveforms of each burst from two intracellular recordings. The coefficient of determination ($R^2$) was

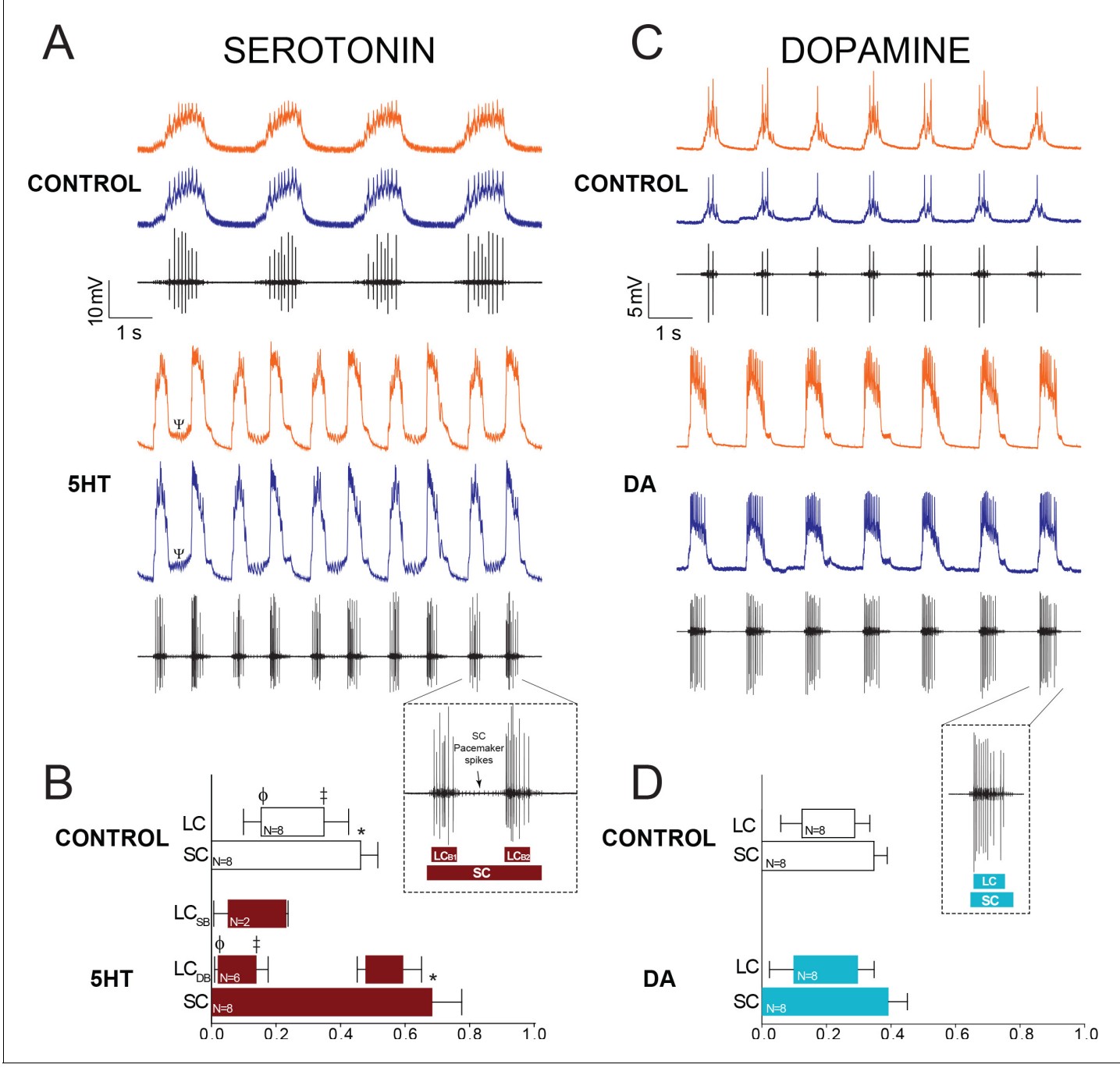

**Figure 2.** Effects of $10^{-6}$M 5HT and $10^{-5}$M DA on output of the CG network. (**A**) Effects of $10^{-6}$M 5HT on the cardiac network. For each condition (Control and 5HT) the top two traces are intracellular recordings from two different LCs in the same network (LC3 and LC4) and the bottom trace is a simultaneous extracellular recording from the CG trunk. These recordings are taken from the same preparation before and after exposure to 5HT. In this case, the preparation exhibited 'double-bursting' output in 5HT, whereby two LC bursts are generated from continuous input from SC pacemaker cells. Note how the LC membrane potential does not return to baseline between two bursts of one full pacemaker cycle (designated by a Ψ in the 5HT intracellular recording). (**B**) Phase relationships of SCs and LCs in 5HT. Phase relationships for double-bursting (LC_DB, N = 6) and single-bursting (LC_SB, N = 2) LCs are shown separately for the 5HT condition. Bars represent mean ±SD. Inset shows how extracellular traces were used to quantify the phase parameters for the SC pacemakers and double-bursts (LC_B1, LC_B2) in 5HT preparations. Significant differences (p<0.01, paired t-tests) relative to control are denoted with the presence of distinct symbols for SC off (*), LC on (φ), and LC off (‡). These were analyzed only for the double-bursting preparations, where statistical power was sufficient. Although double-bursting is a distinct form of output unique to the 5HT condition, we compared the LC on and LC off for the first LC burst to demonstrate a significant phase advance of the initial LC bursting in this condition. (**C**) Effects of $10^{-5}$M DA on the cardiac network. Recordings as in panel A. (**D**) Phase relationships of SCs and LCs in DA (N = 8). DA does not initiate a distinct double-

*Figure 2 continued on next page*

*Figure 2 continued*
bursting output in LCs, and there were no significant changes in phase relationships in DA. Inset shows how extracellular traces were used to quantify the phase parameters for the SC pacemakers and single LC bursts in DA preparations.
DOI: https://doi.org/10.7554/eLife.39368.003

then used to quantify how accurately the voltage of one cell predicts the voltage of the other. This provides a baseline measure of synchrony for each burst and allowed us to track relative changes.

At the onset of serotonergic modulation ($10^{-6}$M), there was an acute reduction in synchrony as measured by $R^2$. *Figure 3A* illustrates typical acute effects of 5HT application. Differences in voltage waveforms appear between LC3 and LC5 after 5HT perfusion. $R^2$ values for every burst during a single experiment are plotted in *Figure 3B* to visualize the typical time course of changes in synchrony. Synchrony reliably reached a minimum within several minutes (mean = 9.1 min) before stabilizing and showing a slow, modest recovery of synchrony. For statistical analysis in Figure 3C, $R^2$ values from 10 consecutive bursts were averaged at three time points: Control (5 min prior to modulation), acute 5HT modulation (sampled at the point of maximum desynchronization), and again after 30 min of exposure to 5HT (N = 8 preparations). Synchrony significantly changed as a result of 5HT exposure (p<0.001, RM ANOVA), resulting in a decrease of synchrony in acute 5HT ($R^2$ = 0.967 ± 0.012 control, 0.893 ± 0.081 acute 5HT; p<0.01, Tukey test). There then was a significant increase in synchrony between acute 5HT and 30 min of continuous 5HT perfusion (0.927 ± 0.066; p<0.01, Tukey test). but synchrony in 30 min of 5HT was still significantly below baseline (p<0.05, Tukey test). During 5HT-induced double-bursting, the first and second of the two bursts typically displayed distinct waveform synchrony values. The $R^2$ values for both the first and second bursts were lower than control, but which of the two bursts in the cycle had lower $R^2$ values varied across preparations.

In stark contrast, burst waveforms for LCs in DA remained completely and strikingly synchronized (*Figure 3D*). The scatterplot in *Figure 3E* shows $R^2$ for each burst in one preparation during a full 30 min of DA perfusion. There were no changes in synchrony during any DA perfusion experiment (*Figure 3E and F*; p=0.784, RM ANOVA, N = 8). The acute data point was sampled at 9.1 min to match the mean time point used for sampling in 5HT.

## DA, but not 5HT, modulates coupling conductance

We hypothesized that modulation of electrical coupling may be responsible for the different effects of 5HT and DA on LC synchrony. One possibility is that modulation with 5HT does not increase the strength of electrical coupling sufficiently to prevent differential effects on cell excitability from resulting in desynchronization. DA may increase coupling strength enough to ensure synchrony.

As an indicator of the strength of electrical coupling, we measured the coupling coefficient (see Materials and methods) between LC3 and LC5 at control and after 15 min of modulation. Coupling coefficients were not significantly different between control (0.043 ± 0.035) and 15 min modulation in 5HT (0.037 ± 0.042) (*Figure 4A*; N = 8). In contrast, DA increased coupling coefficient by 41% (0.035 ± 0.030 in control; 0.049 ± 0.043 in DA) (p<0.05, paired t-test, N = 6; *Figure 4A*). Input resistance was not significantly changed in either 5HT (3.62 ± 4.40 MΩ in control; 3.78 ± 4.11 MΩ in 5HT) or DA (3.85 ± 2.34 MΩ in control; 3.33 ± 1.77 MΩ in DA) (*Figure 4A*). These results suggest that an increase in coupling coefficient in DA may be the result of a direct increase of coupling conductance in the presence of DA.

While the coupling coefficient is a useful description of the functional coupling relationship, it does not identify the electrophysiological mechanism, that is a change in membrane resistance or a change in coupling conductance. LC3 is coupled to LC4 and LC5 in the ganglionic trunk distal to the somata. This electrotonic distance prevents accurate calculation of coupling conductance using somatic recordings on different branches of the ganglion. The branch containing LC4 and LC5 can be isolated from the network by thread ligature creating the ideal conditions for measuring the coupling conductance between these somata (*Figure 4B*). With two electrodes in each cell, we used hyperpolarizing current injections to determine the gap junctional resistance in both directions independent of membrane resistance (see Materials and methods). 5HT had no apparent effect on coupling conductance when measured from LC4 to LC5 (0.490 ± 0.179 µS in control; 0.503 ± 0.186 in 5HT; p=0.329, paired t-test), or from LC5 to LC4 (0.487 ± 0.186µS in control, 0.516 ± 0.211µS in

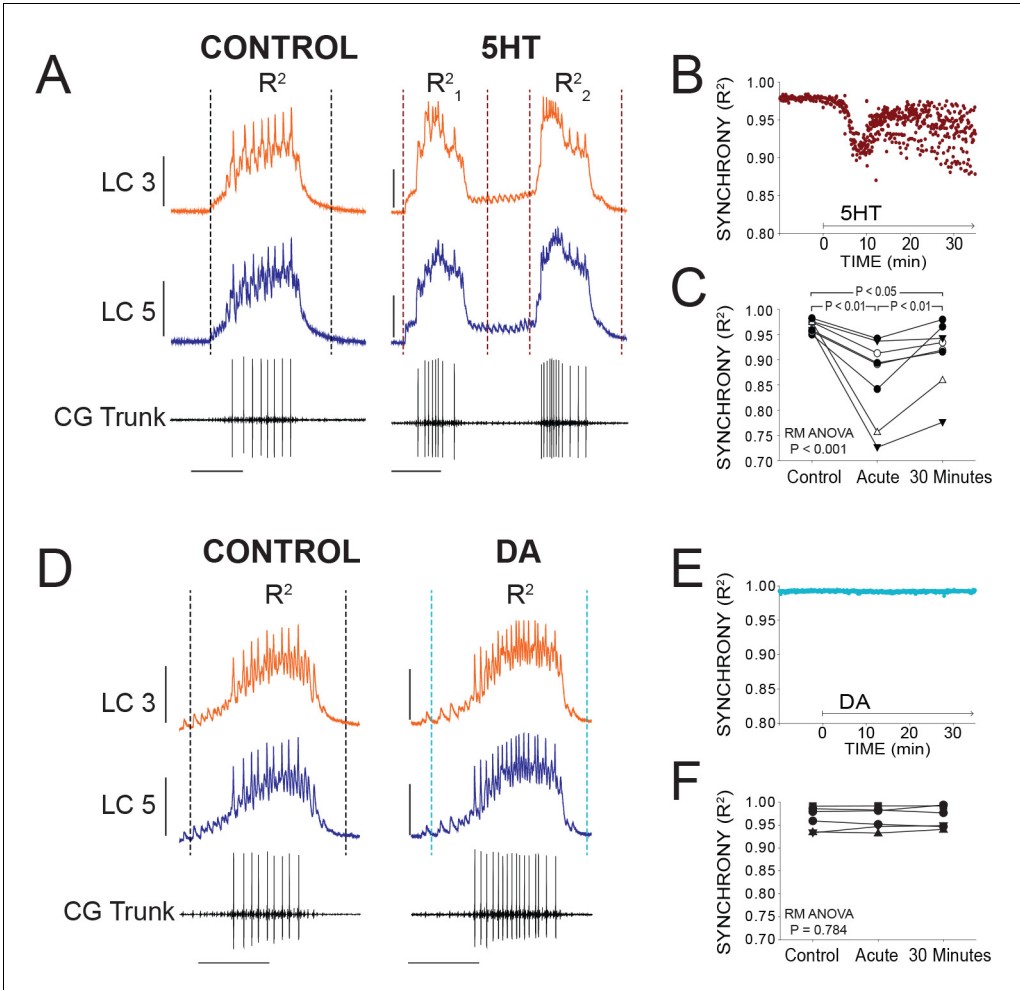

**Figure 3.** Effects of 5HT and DA on Synchrony of LC Voltage Waveforms. (**A**) Representative traces show that LCs with virtually identical control activity produce different burst waveforms after application of 5HT. Double-bursting occurred 6 of 8 preparations in 5HT. Dashed lines designate points between which were used for calculation of $R^2$ values (see Materials and methods). In 5HT double-bursting preparations, each LC burst waveform was treated as distinct for measurements of $R^2$ (i.e. $R^2_1$, $R^2_2$). Scale bars = 10 mV and 1 s. (**B**) Waveform synchrony ($R^2$) was calculated for every burst across a full experiment, and a scatterplot shows the synchrony of bursts for 10 min of control activity followed by 30 min of continuous perfusion of 5HT in one example preparation. An acute loss of synchrony accompanies the onset of modulation. (**C**) $R^2$ was averaged for 10 consecutive bursts at each of 3 time points: control (5 min prior to perfusion), Acute (at the point $R^2$ reached a minimum), and after 30 min of modulation. A one-way Repeated Measures ANOVA indicated that there was a significant effect of 5HT across groups ($p<0.001$, N = 8). Post-hoc testing revealed a significant decrease in $R^2$ from control to acute 5HT ($p<0.01$, Tukey test), and a significant increase between acute and 30 min ($p<0.01$, Tukey test). Synchrony was not restored to control levels after 30 min ($p<0.05$, Tukey test). N = 8 preparations. (**D**) Representative traces show that excitability and network output are affected by DA, but LCs remain synchronized. Dashed lines designate points between which were used for calculation of $R^2$ values. Scale bars = 10 mV and 1 s. (**E**) $R^2$ was calculated for every burst across a full experiment. Scatterplot shows 10 min of control activity followed by 30 min in DA for a representative preparation. $R^2$ values are largely unaffected by changes in activity caused by DA. (**F**) $R^2$ was averaged for 10 consecutive bursts at each of 3 time points: control (5 min prior to perfusion), Acute (9.1 min), and after 30 min of DA. There were no significant effects of DA on synchrony indicated as a result of one-way Repeated Measures ANOVA ($p=0.784$, N = 8).

DOI: https://doi.org/10.7554/eLife.39368.004

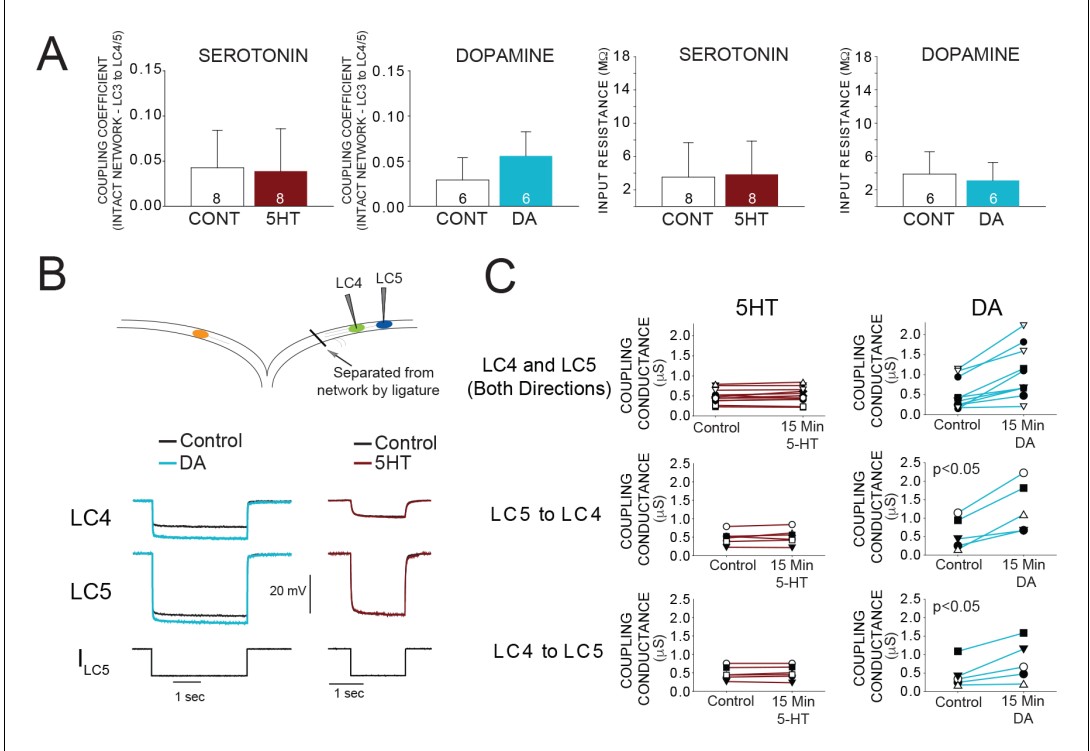

**Figure 4.** Effects of 5HT and DA on Electrical Coupling. (A) Coupling Coefficients measured in the intact network for LC3-LC4 or LC3-LC5 were not significantly changed by 5HT. Mean coupling coefficient increased by 41% in DA (p<0.01, paired t-test). There were no effects of 5HT or DA on cell input resistance. Sample sizes noted in each bar. (B) A reduced preparation where LC4 and LC5 somata were physically isolated from the network by thread ligature was used to test the direct effect of 5HT and DA on coupling conductance. Representative traces of current injections in these isolated pairs of cells that were used to calculate coupling conductance are shown before and after DA exposure (left) and 5HT exposure (right). Traces in black show voltage responses of LC4 (top) and LC5 (middle) to a −8 nA hyperpolarizing current injection into LC5 (bottom). Overlaid traces show the voltage response to the same current injection after DA (left, blue) and 5HT (right, maroon). Control traces in the 5HT recordings are difficult to see due to near complete overlap when 5HT and Control recordings were superimposed. (C) Coupling conductance between LC4 and LC5 was unchanged by 5HT, and increased by DA. Top row in C shows coupling conductance in both directions (i.e. LC4 to LC5, and LC5 to LC4) for N = 5 preparations before and after modulation. These data are then separated by directionality. Coupling Conductance was significantly increased in DA in both directions (p<0.05 for each, mean increase 149%, N = 5).

DOI: https://doi.org/10.7554/eLife.39368.005

5HT; p=0.367, paired t-test). DA significantly increased coupling conductance both from LC4 to LC5 (0.455 ± 0.368µS in control, 0.818 ± 0.552 in DA; p<0.05, paired t-test) and from LC5 to LC4 (0.585 ± 0.435µS in control, 1.294 ± 0.702µS in DA; p<0.05, paired t-test) measured after 15 min of DA exposure (*Figure 4C*).

## Co-application of DA and 5HT prevents desynchronization and induces double bursting

Our previous work demonstrated that a compensatory increase in electrical coupling among LCs rescued synchrony after treatment with the channel blocker TEA (*Lane et al., 2016*). The relatively rapid and large increase in electrical coupling between LCs upon exposure to DA led us to hypothesize that co-modulation with DA might *prevent* desynchronization caused by other modulators. We tested the ability of DA to maintain network synchrony during co-modulation with 5HT using the same perfusion and recording protocol as above, by co-applying DA ($10^{-5}$ M) and 5HT ($10^{-6}$ M).

Individual preparations treated with DA + 5HT sometimes showed a small increase or transient decrease in synchrony, which occurred over the same time scale as seen in preparations exposed to 5HT alone. However, across the full set of preparations (N = 8) there was no significant change in synchrony when DA and 5HT were co-applied (p=0.334, RM ANOVA). 5 out of 8 preparations transitioned to the double-bursting mode seen in 5HT alone (*Figures 2A* and *5A*). However, this time the

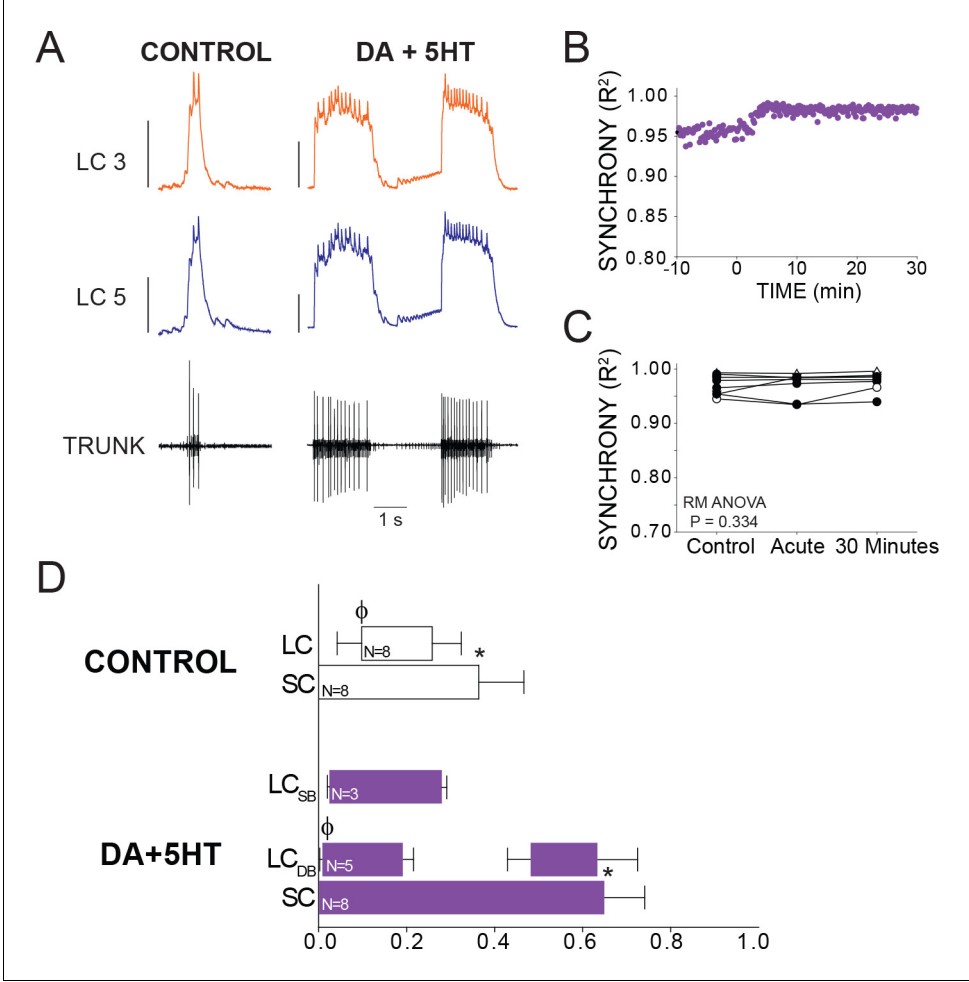

**Figure 5.** Effects of co-application of DA and 5HT on bursting output and synchrony. (**A**) Representative traces show LCs maintain synchronized voltage waveforms during co-application of DA and 5HT. 5 out of 8 preparations transitioned to double-bursting, and network output shows increased number of spikes per burst, spike frequency in each burst, burst duration and LC duty cycle. Scale bars = 10 mV and 1 s. (**B**) $R^2$ was calculated for every burst across a full experiment. Scatterplot shows this for 10 min of control activity followed by 30 min of perfusion with both modulators. (**C**) $R^2$ was averaged for 10 consecutive bursts at each of 3 time points: control (5 min prior to perfusion), Acute, and after 30 min of modulation. There were no significant effects on synchrony detected across groups via one-way Repeated Measures ANOVA (p=0.334, N = 8). (**D**) Phase relationships of SCs and LCs in DA + 5 HT. Phase relationships for double-bursting (LC$_{DB}$, N = 5) and single-bursting (LC$_{SB}$, N = 3) LCs are shown separately. Bars represent mean ±SD. Significant differences (p<0.01, paired t-tests) relative to control are denoted with the presence of distinct symbols for SC off (*) and LC on ($\phi$). These were analyzed only for the double-bursting preparations, where statistical power was sufficient. Although double-bursting is a distinct form of output unique to the 5HT condition, we compared the LC on and LC off for the first LC burst to demonstrate a significant phase advance of the initial LC bursting in this condition.
DOI: https://doi.org/10.7554/eLife.39368.006

double-bursting pattern displayed highly synchronized waveforms (*Figure 5A*). The scatterplot in *Figure 5B* shows $R^2$ for all bursts across a full experiment, in this case revealing a slight increase in synchrony from baseline after the onset of co-modulation. The minimum $R^2$ in 5HT alone occurred an average of 9.1 min after 5HT perfusion began (*Figure 3C*). Because we did not observe a decrease in $R^2$ after perfusion of DA + 5 HT, we performed the $R^2$ analysis for the 'acute' time point 9.1 min after modulator perfusion began in order to best align with the acute time point for 5HT alone. Overall, there was no significant difference in synchrony detected across groups (p=0.334, RM ANOVA, N = 8).

Overall, DA + 5 HT co-modulation elicited changes in network output that were similar to those in 5HT alone. Changes in phase relationships mirrored those seen in 5HT, including significant changes in SC off (p<0.001, paired t-test), and LC on (p=0.011, paired t-test) phase characteristics (*Figure 5D*).

## DA prevents TEA-induced desynchronization

We hypothesize that DA may prevent desynchronization against a variety of perturbations by modulating electrical coupling strength. Our previous study showed that exposing the anterior LCs to the $K^+$ channel blocker TEA produces a substantial loss of LC synchrony as well as a large increase in the number of spikes per burst and spike frequency (*Lane et al., 2016*). For reference, the inset in *Figure 6A* (acute TEA, no DA) includes representative traces of this TEA-induced desynchronization.

If DA can act broadly to maintain synchrony, then we hypothesized that DA should prevent desynchronization in TEA. To test this, a barrier of petroleum jelly was built to protect the posterior (pacemaking) end of the ganglion from the perfusate while leaving anterior LCs exposed (see *Figure 1A*). The anterior LCs were pre-incubated with DA for 5 min, and then the perfusion switched

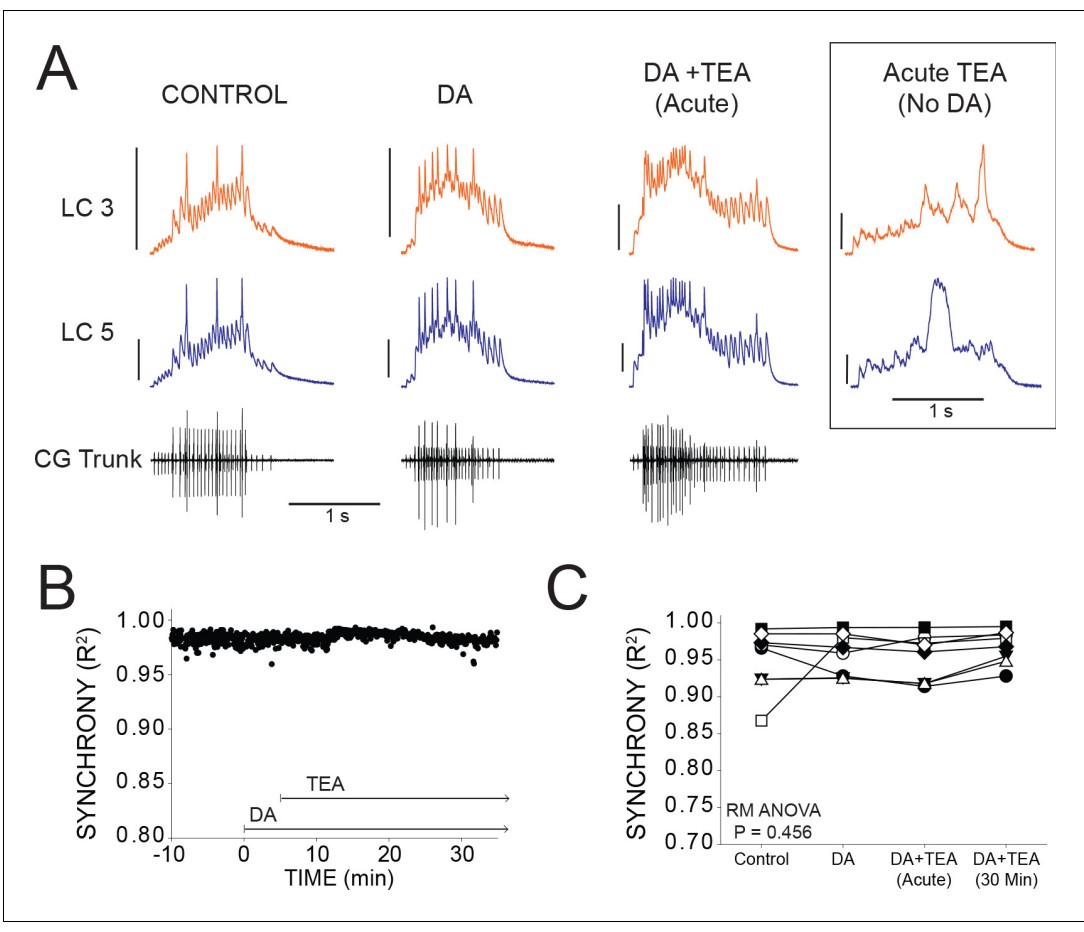

**Figure 6.** DA prevents desynchronization when co-applied with TEA. (**A**) Representative traces for a single preparation in control, DA alone, and DA + TEA. Traces in the box at the far right illustrate acute desynchronization in TEA in the absence of DA (separate preparation). Scale bars = 10 mV and 1 s. (**B**) $R^2$ was calculated for every burst across a full experiment. Scatterplot shows throughout 10 min of control activity followed by 5 min of DA exposure, followed by 30 min in DA and TEA for a single preparation. (**C**) $R^2$ was averaged for 10 consecutive bursts at each of 4 time points: control (5 min prior to perfusion), after 5 min in DA, at the mean time point for desynchronization in TEA observed previously (*Lane et al., 2016*), and after 30 min exposure to the DA+TEA solution. No significant differences were detected across groups via one-way Repeated Measures ANOVA (p=0.456, N = 8).
DOI: https://doi.org/10.7554/eLife.39368.007

to saline containing both DA and TEA. The preparations pre-incubated in DA for 5 min did not show any loss of synchrony as a result of TEA (*Figure 6B and C*). Sample traces (*Figure 6A*) include four time points to show $R^2$ at control (0.950 ± 0.0149), 5 min DA (0.958 ± 0.010), acute exposure to DA + TEA (0.953 ± 0.011), and 30 min exposure to DA + TEA (0.968 ± 0.008). Excitability increased after 5 min with the addition of DA, similar to the increased excitability found after 15 min of DA in previous experiments (*Figure 2D*). After the addition of TEA at 5 min, there was a further increase in the number of spikes per burst (7.89 ± 5.34 in DA and 16.39 ± 7.65 in DA + TEA; p<0.01), spike frequency (8.25 ± 7.95 in DA and 15.19 ± 8.06 in DA + TEA; p<0.01), and cycle period (3.34 ± 0.77 in DA and 3.97 ± 1.06 in DA + TEA; p<0.01). In our previous publication (*Lane et al., 2016*), we observed a substantial and significant decrease in $R^2$ with acute TEA application that occurred an average of 7.5 min after TEA application. In the presence of DA, however, TEA did not cause any detectable change in $R^2$ across groups (p=0.456, RM ANOVA, N = 8). In order to best align our analysis in the present study with the known effects of TEA (*Lane et al., 2016*), we performed the 'Acute' $R^2$ analysis 7.5 min after TEA perfusion began.

## Discussion

Neuromodulation represents a critical mechanism underlying circuit plasticity: by targeting subsets of ionic and synaptic conductances, modulators can reconfigure networks based on environmental feedback to produce entirely distinct circuit outputs leading to behaviorally relevant output that is adapted to the conditions at large (*Harris-Warrick, 2011*; *Nadim and Bucher, 2014*). However, a more recent appreciation of the inherent variability in underlying conductances of individual neurons, even within the same network of the same individual, poses a potential complication for the role of modulation in modifying circuit output. That is, how is reliable neuromodulation (and robust output) achieved when a common modulator targets variable neurons (*Grashow et al., 2009*; *Marder et al., 2014*)? Our study sheds some light on this potential conundrum by revealing complementary roles of the co-modulators 5HT and DA in changing circuit output yet ensuring some features of inherent circuit stability. Specifically, co-application of both modulators leads to a modification of circuit output similar to the effects of 5HT alone, but preserves the synchrony among individual neurons likely through the direct actions of DA on electrical coupling.

### Effects of DA and 5HT on crustacean motor neurons

Both 5HT and DA have been extensively studied in crustacean motor neurons, particularly those of the crustacean stomatogastric ganglion (STG). Generally, these studies have examined the effects of each modulator in isolation, while studies of the combined effects of multiple modulators are rare. In particular, 5HT is known to reduce the conductance of both transient and persistent components of a calcium-dependent $K^+$ current ($I_{KCa}$) in STG cells, an effect that is mimicked by the application of tetraethylammonium [TEA] (*Kiehn and Harris-Warrick, 1992a*). In addition, 5HT enhances a slow voltage-dependent $Ca^{2+}$ current ($I_{Ca}$) in STG cells (*Kiehn and Harris-Warrick, 1992b*; *Zhang and Harris-Warrick, 1995*). These same $I_{KCa}$ and $I_{Ca}$ currents are known to be present in LCs of the cardiac network, and demonstrated to vary over a wide range of conductance magnitudes (*Ransdell et al., 2012*; *Ransdell et al., 2013a*; *Ransdell et al., 2013b*). Furthermore, treatment of LCs with TEA results in an increase in excitability and a loss of network synchrony amongst LCs (*Ransdell et al., 2013a*; *Lane et al., 2016*). In addition, 5HT has only been reported to have very weak or no effect at all on electrical coupling in STG neurons (*Johnson and Harris-Warrick, 1990*). These known features of serotonergic modulation in other crustacean motor neurons are therefore quite consistent with our data in this study. Specifically, 5HT causes an increase in excitability of LCs, as well as causes network desynchronization. We suggest that 5HT modulation has distinct effects on each LC within a ganglion due to the underlying variability of $K^+$ and $Ca^{2+}$ currents in each cell, and yet because 5HT has no effect on coupling this results in distinct hyperexcitable outputs from each cell that manifests as loss of synchronous activity.

DA has distinct and widespread effects on STG cells, both in terms of output and subcellular targets. DA is known to target the transient potassium current $I_A$ (*Kloppenburg et al., 1999*; *Zhang et al., 2010*) as well as $I_H$, $I_{NaP}$, and $I_{Ca}$ in STG cells (*Harris-Warrick and Johnson, 2010*). The influence of DA on each conductance type is dependent on cell type (*Harris-Warrick and Johnson, 2010*; *Zhang et al., 2010*) as well as the concentration and application time course of the modulator

(*Rodgers et al., 2011*; *Rodgers et al., 2013*). Furthermore, DA has been shown to both increase and decrease electrical coupling among STG cells in a cell-type-specific manner (*Johnson et al., 1993*). While $I_H$ is not present in crab cardiac LCs, $I_A$, $I_{NaP}$, and $I_{Ca}$ have all been identified and shown to vary over a wide range of conductance magnitudes in these cells (*Ransdell et al., 2012*; *Ransdell et al., 2013a*). Pharmacological blockade of $I_A$ in LCs also has been shown also to induce hyperexcitability and loss of synchrony in cardiac network output (*Ransdell et al., 2012*). Therefore, our prediction was that we should see similar loss of synchrony with DA application in this study. This was clearly not the case – while our data demonstrate an increase in excitability of LCs exposed to DA, they never experience any decrease in synchrony. It was only when we determined that DA also directly increases electrical coupling that we could propose a mechanism for the maintenance of synchrony despite the targeting of variable underlying conductances across LCs in DA. Namely, that DA's influence on coupling prevents desynchronization and is effectively protective against the potentially variable impacts of modulation across LCs within the network. Although the network effects of individual modulators have typically been examined in isolation, this provided a context in which to interpret the combined effects of multiple modulators. This hypothesis was tested by co-applying DA and 5HT and by co-applying DA and TEA. The fact that neither of these conditions resulted in loss of synchrony is consistent with the hypothesis of a protective effect of increased coupling via DA in the cardiac network.

## Co-application of DA and 5HT has additive complementary effects on network output

While we directly tested whether DA is able to preserve network synchrony in the face of divergent effects on variable LCs within a ganglion through co-application with both TEA and 5HT, we had no prediction as to what the combined influence of 5HT and DA would be on the cardiac network output overall. It is well established that individual biogenic amine modulators have distinct effects on both the CG (*Miller et al., 1984*; *Berlind, 1998*; *Berlind, 2001*; *Cruz-Bermúdez and Marder, 2007*) and the STG (*Flamm and Harris-Warrick, 1986a*; *Flamm and Harris-Warrick, 1986b*). So much so that distinct output modes can be attributed to DA, 5HT, and octopamine individually (*Flamm and Harris-Warrick, 1986a*), and is thought to underlie some of the diversity of circuit reorganization and outputs that has been demonstrated in these networks previously. Furthermore, few co-modulation experiments have been performed that specifically determine whether modulator effects will enhance, occlude, or have distinct effects from one another. Therefore, we did not anticipate that DA and 5HT would have largely complementary, and perhaps even synergistic effects on the ganglion. Our data demonstrate when these modulators were co-applied, the network output took on the hallmarks of 5HT modulation – including double-bursting of the LCs, enhanced excitability, and changes in spike frequency and bursting output – yet were protected from the loss of synchronous activity that accompanies 5HT-only application. These results suggest that DA and 5HT in these cells have distinct signaling pathways, subcellular targets, and ultimately result in a more linear combination of modulator effects than occlusion or saturation (*Li et al., 2018*). It remains to be seen whether DA interacts in this fashion with other modulators, including neuropeptides (*Miller and Sullivan, 1981*; *Cruz-Bermúdez and Marder, 2007*). However, our results indicate that DA acts to ensure robust activity through the coordinated, synchronous action of all 5 LC motor neurons.

Only three pairs of axons provide extrinsic innervation of the CG – one pair is dopaminergic, the others are cholinergic and GABAergic (*Delgado et al., 2000*; *Cooke, 2002*). The dopaminergic fibers have many synaptic connections on anterior LCs, their neuropil (in the vicinity of the sites of electrical coupling), and the posterior of the ganglion near the pacemaker cells (*Fort et al., 2004*). This projection pattern is well situated to provide a potential means for direct delivery of DA to the CG over the immediate time scales of neuromodulator release, in addition to hormonal exposure to DA, through fibers that are rapidly responsive to physiologically relevant stimuli (*Maynard, 1953*; *Guirguis and Wilkens, 1995*; *Jury and Watson, 2000*; *Fort et al., 2004*). This would potentially allow for feedback transmitted through dopaminergic fibers to rapidly 'prime' the network for subsequent modulation delivered hormonally, thus preventing desynchronization as a result.

## Conclusion: Neuromodulation, Electrical Coupling, and Robustness of Network Output

While neuromodulation is largely a phenomenon associated with network plasticity and changes in network output, a more recent appreciation of the additional role of neuromodulation in network robustness has begun to emerge. It has been proposed that the somewhat diffuse and often opposing actions of modulators on different components of the same circuit may act to stabilize the modulated state of networks, and preventing 'overmodulation' that could render a network non-functional or pathological (*Harris-Warrick and Johnson, 2010*; *Marder, 2012*). For example, 5HT increases the set of intrinsic and synaptic current magnitude combinations that can give rise to specific behaviors such as bursting in a half center oscillator (*Grashow et al., 2009*), ensuring a larger parameter space over which appropriate output can be generated in the context of multiple modulation and hence increasing robustness of output. More recently it has been shown that neuromodulation can compensate for temperature-induced loss of motor pattern output in the STG (*Städele et al., 2015*), thus acting in a neuroprotective manner. Electrical coupling itself, as well as modulation of coupling interactions, also play a role in circuit robustness, allowing for distributed network solutions to maximize output consistency under different conditions or with variable underlying physiology of network constituents (*Kepler et al., 1990*; *Gutierrez and Marder, 2013*; *Marder et al., 2017*).

Modulation of coupling via DA may have convergent effects on such robustness across different taxa, as modulation of electrical synapses by DA is known to be involved in many circuits and species. DA directly modulates gap junctional conductance and network activity in horizontal cells (*Piccolino et al., 1984*; *He et al., 2000*), AII amacrine cells (*Kothmann et al., 2009*), and rod cells (*Jin et al., 2015*) of the retina. DA modulation of coupling conductance is also involved in sensorimotor function during copulation in *C. elegans* (*Correa et al., 2015*). Furthermore DA has been shown capable of modulating the electrical component of the mixed electrical-chemical synapses of auditory afferents onto the fish Mauthner cell (*Pereda et al., 1992*; *Cachope et al., 2007*; *Cachope and Pereda, 2012*). While electrical coupling is known to support synchronized activity in many systems, and DA is known to modulate such coupling across taxa, to our knowledge this is the first study that directly implicates the neuromodulation of gap junctional conductance in directly counteracting desynchronizing perturbations to enhance network robustness. Furthermore, DA can do so in this system and maintain the ability of other modulators, in this case 5HT, to alter output in an independent fashion that is protected from potentially detrimental or destabilizing alterations to coordinated network activity. This expands the understanding of how neuromodulators can interact to ensure appropriate flexibility in circuits to respond to changes in sensory feedback and produce appropriate context-specific output, but ensure that overall robustness of the network is maintained within an acceptable parameter space.

# Materials and methods

**Key resources table**

| Reagent type (species) or resource | Designation | Source or reference | Identifiers | Additional information |
| --- | --- | --- | --- | --- |
| Chemical compound, drug | 5HT | Sigma | H7752 | 10−6M |
| Chemical compound, drug | DA | Acros Organics | 122000100 | 10−5M |
| Chemical compound, drug | TEA | Acros Organics | 150901000 | 10−5M |
| Software, algorithm | Phaseburst | http://stg.rutgers.edu/Resources.html | | Spike2 analysis script, provided by Dr. Dirk Bucher, New Jersey Institute of Technology |

## Animals

Adult male Jonah crabs, *Cancer borealis*, were purchased and shipped overnight from The Fresh Lobster Company (Gloucester, MA). Crabs were maintained in artificial seawater at 12°C until used.

Crabs were anesthetized by keeping them on ice for 30 min prior to dissection. The complete cardiac ganglion was dissected from the animal and pinned out in a Sylgard-lined petri dish in chilled physiological saline (440 mM NaCl, 26 mM MgCl₂, 13 mM CaCl₂, 11 mM KCl, and 10 mM HEPES, pH 7.4–7.5, 12°C). Chemicals were obtained from Fisher Scientific unless otherwise noted.

## Electrophysiology

The CG network is comprised of 9 cells: 4 Small Cell (SC) pacemaker interneurons which give excitatory input to 5 Large Cell (LC) motor neurons. Stainless steel pin electrodes were connected to differential AC Amplifier (A-M Systems model 1700) for extracellular recording. One pin was placed inside a petroleum jelly well built around the ganglionic trunk and the other placed in the bath outside the well (*Figure 1A*). The ganglionic trunk contains axons from all 9 cells, and thus serves to monitor the spiking output of the entire network. During normal activity, LCs produce consistent levels of rhythmic output and pairs of LCs show nearly identical voltage waveforms. LC and pacemaker spikes are easily distinguishable by their relative amplitudes, and by intracellular potential changes in the LCs (*Figure 1B*).

LC somata are easily visible within the nerve and can be individually desheathed for intracellular sharp electrode recordings. Intracellular sharp electrodes containing 3M KCl (8–25 Mç) were used to simultaneously monitor the voltage activity in the somata of two anterior LCs. All paired intracellular recordings were from LC3 and either LC4 or LC5. Amplifiers from Axon Instruments were used (AxoClamp 900A, MultiClamp 700B, AxoClamp 2B). Current clamp protocols were created and run using Clampex 10.3 software (Molecular Devices). Electrical coupling in the intact network was measured by hyperpolarizing current injection (1–6 nA) when LCs reached resting membrane potential between bursts. Cells were injected one at a time while measuring voltage changes in both cells. Coupling coefficients were calculated as the ratio: ($\Delta V_{coupled\ cell}$ / $\Delta V_{Injected\ Cell}$). For both amines, the coupling coefficient at control was compared to the measured value after 15 min of modulation. Changes in coupling coefficient could ultimately be influenced by two fundamentally different mechanisms: altered conductance of the non-junctional membrane, or modification of gap junctional conductance. LC3 and LC5 are coupled to one another and the rest of the network distal to the site of our recordings, making calculation of coupling conductance between these two cells problematic. Our reduced preparation of the thread ligatured LC4/LC5 branch provides an electrotonically compact two-cell preparation in which we could calculate coupling conductance. To calculate electrical coupling conductance, we followed methods outlined by Bennett 1966. Briefly, it assumes two compartments and applies the equation (expressed as in *Haas et al., 2011*):

$$R_c = \frac{R_{in,1} * R_{in,2} - R_{(12)}^2}{R_{(12)}}$$

where $R_c$ represents the coupling resistance, $R_{in,1}$ represents the apparent input resistance of Cell 1, $R_{in,2}$ represents the apparent input resistance of Cell 2, and $R_{12}$ represents the transfer resistance from Cell one to Cell 2. Coupling conductance is then the inverse of coupling resistance $G_c = 1/R_c$. Apparent input resistance and transfer resistance were calculated from alternately injecting hyperpolarizing current steps (−2 to −8 nA) into LC4 and LC5.

## Modulator application

Serotonin (5HT; Sigma), dopamine (DA; Acros Organics), and tetraethylammonium (TEA; Acros Organics) were maintained as concentrated stock solutions and diluted to final concentrations in physiological saline. Whenever present, concentrations were as follows: $10^{-6}$ M 5HT, $10^{-5}$ M DA, 25 mM TEA. All perfusions occurred at a rate of approximately 2 ml/min. Solutions were pre-chilled to maintain a constant temperature of 12°C in the dish.

LC variability can be exploited by applying the K⁺ channel blocker TEA exclusively to anterior LCs to desynchronize their activity (*Lane et al., 2016*). To test whether neuromodulation could desynchronize LC activity, we considered perfusion of neuromodulators over the entire network to be a more biologically relevant challenge to test synchrony. There is no reason to suspect any biological conditions in which only the anterior LCs would be exposed to modulators. In vivo, the entire CG is exposed to both serotonin (5HT) and dopamine (DA) as hormonal modulators released from the pericardial organ or other neurohormonal sites (*Cooke, 1966*; *Fort et al., 2004*; *Maynard and*

*Welsh, 1959*). In addition, a pair of extrinsic dopaminergic fibers innervates the CG from the thoracic ganglion and forms abundant synaptic contacts in both the anterior and posterior regions of the ganglion which provides a rapid and direct route for dopaminergic modulation (*Cooke, 2002*; *Fort et al., 2004*). Baseline activity was measured during a sham perfusion of physiological saline before the source of the perfusion was switched to saline containing neuromodulators and/or channel blockers. Simultaneous intracellular recordings monitored somatic burst potentials from multiple anterior LCs, and an extracellular well on the ganglionic trunk monitored activity of the entire network.

In some experiments, a petroleum jelly well was used to protect the posterior end of the ganglion from the perfusate to selectively expose the anterior LCs to 5HT, DA, or DA + TEA (*Figure 1A*). For experiments in which TEA was applied only to the anterior portion of the network, whole-network modulation was achieved by adding DA to the posterior end of the ganglion by pipette at the same time the DA perfusion began.

## Data analysis

Recordings were analyzed using Clampfit 10.3 (Molecular Devices) and Spike 2.9 (CED, Cambridge, UK) software. Statistical analyses were performed using SigmaPlot 11.0. All data are expressed as mean ±SD unless otherwise stated.

Changes in LC burst characteristics, coupling coefficients, and coupling conductance were analyzed with paired *t*-tests after normality testing with Shapiro-Wilk tests. All comparisons for measures of network output between control and modulation are after 15 min of modulation. Each burst characteristic quantified was averaged for 10 consecutive cycles.

Phase data were generated using the 'phaseburst' analysis script in Spike 2 (D. Bucher; available at http://stg.rutgers.edu/Resources.html). Each cycle was considered to begin with the start of a pacemaker burst and end at the start of the next pacemaker burst. The 'On' phase for each LC burst was then calculated as: the delay time between the start of the cycle and the start of the LC burst, divided by the full cycle period. Similarly, the 'Off' phase for each burst (pacemaker and LC) was calculated as: the delay time between the start of the cycle and the end of the burst, divided by the full cycle period. For each individual preparation, phasing data from 10 consecutive cycles were averaged at control, and again after 15 min of modulation. Changes in phase characteristics (SC off, LC on, and LC off) were confirmed to be of normal distribution via Shapiro-Wilk tests, and subsequently analyzed with paired *t*-tests.

R-values (and thus $R^2$) as measures of synchrony were obtained by Pearson correlation tests as described previously (*Lane et al., 2016*; *Ransdell et al., 2013a*). Boundaries were set around the intracellular burst waveforms for cross-correlation ($R^2$) analysis using Spike 2 Version 9 (Cambridge Electronic Design) as follows: Pacemaker spikes begin before LC spikes, produce extracellular spikes of much smaller amplitude, and cause visible EPSPs in LCs that drive LC burst potentials. Thus, the time stamp of the first extracellular pacemaker spike in each burst reliably marked the beginning of the intracellular burst waveform used for cross-correlation, and also marked the resting membrane potential immediately before the beginning of the LC burst waveform. The intracellular burst waveform was considered to end upon return to resting membrane potential.

However, when 5HT caused 2 LC bursts per pacemaker cycle (i.e 'double-bursting'), it was necessary to modify this procedure slightly because LCs did not return to resting membrane potential between the first and second burst of the cycle (*Figure 2A*). The beginning of the first LC burst and the end of the second LC burst could still be marked as described above. We measured the delay time required to reach resting membrane potential after the last spike of the second LC burst. We then applied this time increment to establish a comparable end-point for the first LC burst. We next measured the amount of time between the first pacemaker spike of the cycle and the first spike of the first LC burst. This time increment was used to set a comparable beginning time point for the second LC burst for cross-correlation. Analysis of waveform synchrony where more than two experimental groups or time points were compared was performed with one-way Repeated Measures ANOVA (RM ANOVA) after data were confirmed to be of normal distribution by Shapiro-Wilk tests. If a significant ANOVA effect was detected, pairwise testing was done with post-hoc Tukey tests.

The sample sizes to compare waveform synchrony were calculated with power analyses based on data reported in two previous studies with highly similar experimental manipulations of exposure of LCs to compounds that cause loss of synchrony, which yielded target sample size of N = 6–8 to yield

a power ≥0.8. Sample sizes for changes in network output following modulator exposure were based on similar data in our previous work (*Ransdell et al., 2012*), and yielded target sample sizes of N = 5–6 to achieve a power ≥0.8. Power analyses were conducted based on the use of paired *t*-tests to analyze the data. All sample sizes used in our studies are reported in Figure Legends and/or in the Results section when significance values are reported.

## Additional information

### Funding

| Funder | Grant reference number | Author |
|---|---|---|
| National Institutes of Health | R01MH046742-29 | David J Schulz |

The funders had no role in study design, data collection and interpretation, or the decision to submit the work for publication.

### Author contributions

Brian J Lane, Conceptualization, Data curation, Formal analysis, Investigation, Methodology, Writing—original draft; Daniel R Kick, Resources, Software, Formal analysis, Methodology, Writing—review and editing; David K Wilson, Data curation, Formal analysis, Investigation; Satish S Nair, Conceptualization, Resources, Writing—review and editing; David J Schulz, Conceptualization, Data curation, Supervision, Funding acquisition, Investigation, Methodology, Project administration, Writing—review and editing

### Author ORCIDs

Brian J Lane http://orcid.org/0000-0003-3416-2377
Daniel R Kick http://orcid.org/0000-0002-9002-1862
Satish S Nair http://orcid.org/0000-0002-1489-7029
David J Schulz http://orcid.org/0000-0003-4532-5362

### Decision letter and Author response

Decision letter https://doi.org/10.7554/eLife.39368.011
Author response https://doi.org/10.7554/eLife.39368.012

## Additional files

### Supplementary files

• Transparent reporting form
DOI: https://doi.org/10.7554/eLife.39368.008

### Data availability

All data generated or analyzed during this study are included in the manuscript and supporting files.

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
