## [Decision Letter]

Thank you for submitting your article "Dopamine maintains network synchrony via direct modulation of gap junctions in the crustacean cardiac ganglion" for consideration by *eLife*. Your article has been reviewed by three peer reviewers, including Ronald L Calabrese as the Reviewing Editor and Reviewer #1, and the evaluation has been overseen by a Senior Editor. The following individual involved in review of your submission has agreed to reveal their identity: Carlos D Aizenman (Reviewer #2).

The reviewers have discussed the reviews with one another and the Reviewing Editor has drafted this decision to help you prepare a revised submission.

Summary:

In this manuscript, the authors investigate Large Cell motor neurons of the crab (*C. borealis*) cardiac ganglion, which are known to have variable membrane conductance magnitudes even within the same individual, yet produce identical synchronized activity in the intact network. In their previous study (Lane et al., 2016) they blocked a subset of K^+^ conductances across the Large Cells, resulting in acute loss of synchronous activity. Here they tested the hypothesis that this same variability of conductances could make LCs vulnerable to desynchronization during neuromodulation. They compared the exposure of the Large Cells to serotonin (5HT) and dopamine (DA) while recording simultaneously from two Large Cells and extracellularly from the motor network. Both amines had distinct excitatory effects on Large Cell output, but only 5HT caused desynchronized output. They show that DA rapidly increased gap junctional conductance and that co-application of both amines induced 5HT-like output, but waveforms remained synchronized. DA also prevented desynchronization induced by the K^+^ channel blocker tetraethylammonium (TEA). These results suggest that modulation of electrical coupling by DA plays a protective role in maintaining network synchrony to make this key feature of the system more robust to the modulation of excitability by other substances.

The writing is generally clear and necessary data is presented. This is a significant advance because it shows that the neuromodulation of electrical coupling directly counteracts desynchronizing perturbations to enhance network robustness, similar to the previous study where homeostatic regulation of electrical coupling restores synchrony after pharmacological perturbation. It should be of wide interest.

Essential revisions:

1) There was a general concern that how the analysis of double bursts was performed is not clear. Bursts and period need to be clearly defined. Please refer to the comments of reviewer 1 and 3.

2) The statistical analysis needs to be better justified and explained. Please see the comments of reviewer 3.

Reviewer #1

1) The authors do not do a good job explaining how burst characteristics are measured (the synchrony measure is a bit better explained), which caused this reviewer to be confused especially when 'double bursting' occurs.

The authors state "Intracellular burst waveforms were considered to begin with the 1st EPSP from pacemaker activity and ended upon return to resting membrane potential." These points of beginning and end are not indicated on any of the intracellular traces making the reviewer struggle. Moreover, they clearly are not used to determine bursts in this study because as far as I can see each part of the 'double burst' is considered and individual burst in the synchrony analyses. How were burst determined during double bursting for synchrony analyses? Mark the bursts on the intracellular traces.

Next the authors must indicate explicitly how burst characteristics were determined from the extracellular traces. What is a burst and how is it determined during single and double bursting? Mark beginning and end times on extracellular traces. What is a cycle when measuring during double bursting? Clearly it is not a pacemaker cycle, which from a physiological point of view is a heartbeat cycle.

Explicitly define what a cycle is when measuring during single and then double bursting. I suspect it is start (first spike) of burst to start of burst and represents a motor pattern cycle not necessarily a pacemaker cycle. During double bursting are there not two different cycles (and thus two different periods) going on simultaneously, one between the two 'bursts' of a given double burst and one from the beginning of the second 'burst' of a double burst to the beginning of the first 'burst' of the next double burst? Are these two different periods different? One is an intrinsic motor neuron period and one a pacemaker period? What happens when a prep that should be double bursting in 5HT does not? Doesn't this throw composite data off and make a hash?

Please define a burst and a cycle and mark the definitions clearly on the traces.

When I look at the extracellular traces I see a number of different spike sizes in each 'burst'. Which of the different spikes are used in measuring burst duration and spike number and spike frequency. Please mark all counted spikes on the traces. Again, what happens when a prep that should be double bursting in 5HT does not? Doesn't this throw composite data off (e.g., for spike number) and make a hash?

Reviewer #3:

1) Please explain "doublets". For those not familiar with this system, it is not clear that there are 2 bursts per cycle, as the intracellular recordings are bursting in time with the extracellular recording. But there are very small pacemaker spikes in the extracellular recording? Please describe. It would be useful to mark the duration of the pacemaker burst (do we see this as PSPs in the LC neurons?). Also, please describe the "two distinct bursts". Perhaps it would help to include an inset of an expanded time scale of the bursts to highlight their distinctions and their timing relative to the pacemaker activity.

2) It is not clear to me why paired t-test and signed rank tests were used for all statistical analyses in this study. For many of them, there are more than two groups to be compared, why were multiple paired tests performed instead of an ANOVA followed by post-hoc pairwise comparisons? For example (subsection “5HT Desynchronizes Burst Waveforms but DA Does Not**”**), why not use an ANOVA test to compare control, acute, and 30 minutes of exposure? Other examples include the synchrony between 0, 10, and 30 minutes of modulator application (Figure 3), and the comparison of control, DA, and DA+TEA at two time points (subsection “DA Prevents TEA Induced Desynchronization**”**; Figure 5).

---

## [Author Response]

Essential revisions:1) There was a general concern that how the analysis of double bursts was performed is not clear. Bursts and period need to be clearly defined. Please refer to the comments of reviewer 1 and 3.

We struggled as well with the “double bursts” and their quantification, and clearly when we submitted the manuscript we did not succeed in finding a good way to represent this output. So we have thoroughly revised the manuscript with this in mind. In particular, we have provided much more detail about the double-bursting output, including figure changes in Figures 2, 3, and 5 (in particular 2 and 3) that hopefully make all of this much clearer. In essence, we realized why we were struggling was due to the fact that 5HT double-bursting is a distinct output from control. Trying to directly compare burst characteristics of two different network outputs was a mistake that we think led to the majority of the confusion. Furthermore, this manuscript has very little to do with the quantitative effects of 5HT on burst output in this system, which has already been described in more detail for these modulators and several others in a very nice paper by Cruz-Bermudez et al. in 2007 (in the reference list), and in lobsters by Allan Berlind (also cited). By devoting as much figure space as we did to flawed comparisons of burst characteristics between two modes of firing, we distracted readers from the core hypothesis of this work: network synchrony.

To alleviate this confusion, we have switched the reporting of these results to phase analyses instead of burst characteristics. We feel that phase analysis (now seen in Figure 2 and Figure 5) gives the reader a much better feel for how the output of this network is reconfigured by 5HT, provides a basis for a statistical comparison to demonstrate that 5HT is indeed strongly affecting this network, and avoids the “apples and oranges” issue of comparing one burst (control) to two bursts (5HT). Additionally, when appropriate (i.e. in conditions where double-bursting does not occur), we do report burst statistics in the text for anyone interested in knowing how DA and focal application of 5HT to LCs influences these LC burst outputs. We hope the reviewers feel this is a superior way to illustrate the salient points we are trying to make.

2) The statistical analysis needs to be better justified and explained. Please see the comments of reviewer 3.

As a result of the prompting of the reviewers, we have reanalyzed the entirety of the data in the manuscript with Repeated Measures ANOVA followed by post-hoc pairwise Tukey tests when appropriate (i.e. a significant effect was found). This greatly streamlines the analysis, it is the more appropriate analysis, and overall makes the data much easier to grasp. The new format of the analysis is provided throughout, where RM ANOVA results are reported on each figure where appropriate, and when significance was detected the post-hoc comparisons also reported. ANOVA results and sample sizes are included throughout the manuscript.